# Performance Verification of a Flexible Vibration Monitoring System

**Patrick Bointon [1],\*** , **Luke Todhunter [1]** , **Adam Clare [2] and Richard Leach [1]**

[1]   Manufacturing Metrology Team, Faculty of Engineering, University of Nottingham,
   Nottingham NG7 2RD, UK; luke.todhunter@nottingham.ac.uk (L.T.); richard.leach@nottingham.ac.uk (R.L.)
[2]   Advanced Component Engineering Laboratory, Faculty of Engineering, University of Nottingham,
   Nottingham NG7 2RD, UK; adam.clare@nottingham.ac.uk
\*   Correspondence: Patrick.Bointon@nottingham.ac.uk

**Abstract:** The performance of measurement or manufacturing systems in high-precision applications is dependent upon the dynamics of the system, as vibration can be a significant contributor to the measurement uncertainty and process variability. Technologies making use of accelerometers and laser vibrometers are available to rapidly measure and process structural dynamic data but the software infrastructure is yet to be available in an open source or standardised format to allow rapid inter-platform use. In this paper, we present a novel condition monitoring system, which uses commercially available accelerometers in combination with a control-monitoring infrastructure to allow for the appraisal of the performance of a measurement or manufacturing system. A field-programmable gate array (FPGA)-based control system is implemented for high-speed data acquisition and signal processing of six triaxial accelerometers, with a frequency range of 1 Hz to 6000 Hz, a sensitivity of 102.5 mV/ms$^{-2}$ and a maximum sample rate of 12,800 samples per second per channel. The system includes two methods of operation: real-time performance monitoring and detailed measurement/manufacturing verification. A lathe condition monitoring investigation is undertaken to demonstrate the utility of this system and acquire typical machining performance parameters in order to monitor the "health" of the system.

**Keywords:** process monitoring; vibration detection; metrology

## 1. Introduction

The importance of effective industrial measurement solutions to provide accurate and traceable parts is widely acknowledged [1,2]. Therefore, it is vital for the continued progress of rapidly emerging manufacturing technologies, such as additive manufacturing, that we continue to develop new and improved measurement technologies in order to meet the increasing measurement demands [3]. One method to increase confidence in a measuring instrument is to more closely monitor the measurement process, to provide the user with metrics to show the system is performing as specified. Therefore, new methods must be developed to monitor the performance of measurement systems during both the measurement, and check the instrument mechanical performance over time.

Condition monitoring [4] is a technique which is particularly prevalent in machine monitoring applications within the automotive [5,6], aeronautical [7], and manufacturing [8] industries. Condition monitoring is used as an effective tool to promote a predictive maintenance strategy, rather than operating the traditional run-to-break strategy [9], which can lead to catastrophic failures. Structural vibration is a common measure and used in structural condition monitoring, and allows for the detection of faults within the system. In addition, several authors have demonstrated the importance of multipoint condition monitoring [10–12], which allows a system to be more closely monitored at

a number of key areas to enhance the understanding of the system's performance, and improve the ability to easily detect faults within the system and find the location of the root cause.

In machine cutting applications, the tool condition monitoring is most widely performed using accelerometers, as they offer a practical solution which can be readily mounted to the system. However, accelerometers are only capable of measuring the tools dynamics indirectly, and so a noncontact condition monitoring system, capable of directly measuring the tool's dynamic displacement during the machine cutting process remains desirable. A number of authors have demonstrated the promise of noncontact measurement techniques, such as Tatar and Gren [13], Miyaguchi et al. [14] and Wojciechowski [15,16]. Tatar and Gren demonstrated the use of a laser vibrometer to monitor the tool vibrations, radial misalignment and out-of-roundness during the cutting process of a milling tool. Miyaguchi et al. used a capacitive gap sensor (dynamometer) to measure the displacement of the ball end mill during milling cutting process of hardened steel. Wojciechowski used a laser displacement sensor to investigate the quantitative influence of cutting forces and cutter displacements on surface parameters Ra and Rz in milling cutting operations. Despite the promise of these methods, they were performed on milling machines which allow for sufficient space to enable mounting of the required measurement instruments with line-of-sight and without risk of being damaged by the swarf produced during the cutting process. In the case of a lathe, space is limited and any attempt to mount laser measurement instruments inside the cutting chamber is likely to cause damage. In addition, each of these approaches requires expensive equipment and have been used to obtain a single-point measurement. Expanding these approaches to multipoint condition monitoring is cost prohibitive, and so the use of accelerometers remains the best solution at this stage.

Although measurement systems are not prone to catastrophic failures, they are reliant on reliable mechanical performance to provide accurate measurements, and can be sensitive to vibration [17]. In more vibration sensitive systems, the quality of the measurement is dependent upon the dynamic state of the apparatus and so it can be vital that this performance is measured. Despite the importance of the mechanical performance, current measurement systems implement little or no active condition monitoring, and so it is difficult to verify the performance of the system and check for any structural deterioration over time. Taking performance measurements would be beneficial as it would provide an effective condition monitoring tool to verify the systems is performing as stipulated by the manufacturer and measurement standards [18,19]. A measurement report, which included the complimentary characterisation information of the system during the measurement, along with the measurement data itself, would increase confidence in the reliability and accuracy of the measurement data and the system, and the measurement data itself. The review presented here has shown that there are clear gaps in the literature in relation to live characterisation of measurement systems, and so we have developed a novel condition monitoring system which is capable of multipoint three axis condition monitoring and real-time data capture and analysis. The system uses commercially available accelerometers in combination with a control-monitoring infrastructure to allow for the live appraisal of the performance of measurement mechanical structural elements, such as the camera and projector mounting tripod in a fringe projection system or the measurement stage of a focus variation system.

## 2. System Overview

### 2.1. Software

The system (see Figure 1) was developed in LabVIEW, which is a graphical programming environment widely used for systems development as it allows for flexible data acquisition, data analysis and instrument control. In addition, the system also utilised LabVIEW's field programmable gate array (FPGA), real-time and MathScript specialised add-on modules to support its infrastructure. The FPGA hardware embedded in the Compact RIO was configured to create an integrated circuit that can interface with the microprocessor of the Compact RIO and the vibration I/O modules installed, to collect and stream data from the accelerometers. Parallel collection and streaming of data on FPGAs

can operate at higher speeds than processors that perform tasks sequentially. The real-time module was used to ensure accurate timing within our condition monitoring application, which is vital for a stand-alone, multiple channel and high throughput data collection system.

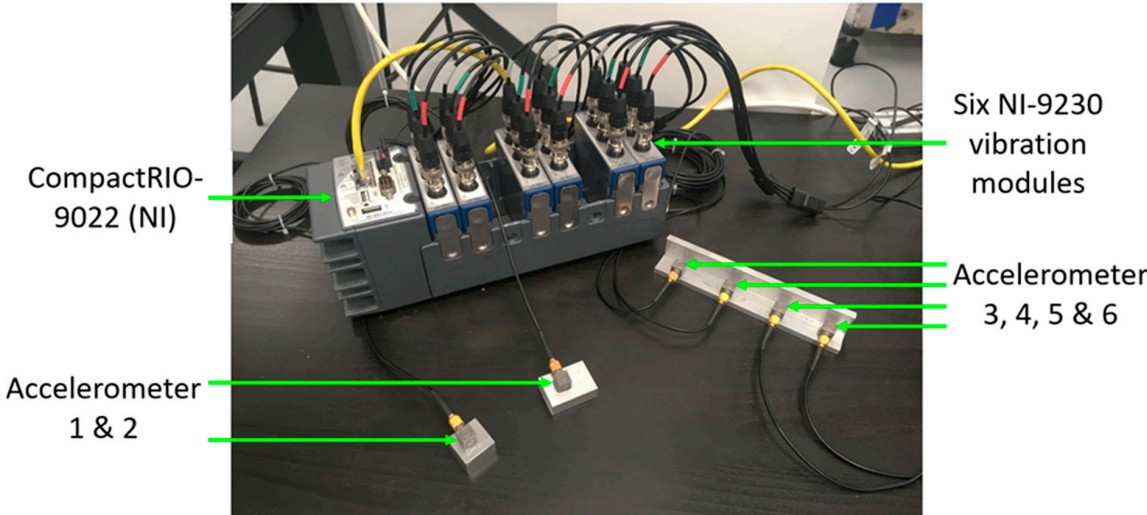

**Figure 1.** The developed multi-accelerometer live monitoring system, which includes six triaxial IEPE accelerometers, a CompactRIO-9022 controller, with an eight-module slot chassis attached and six NI-9230 vibration and sound modules integrated.

## 2.2. Hardware

The system was developed using a National Instruments (NI) CompactRIO-9022 (cRIO) real-time embedded controller (533 MHz CPU, 256 MB DRAM, 2 GB storage), which has a variety of connectivity ports, including two Ethernet, one USB and one serial port. The cRIO was combined with a compatible cRIO chassis, which allows for up to eight C-series I/O modules to be integrated. Six NI-9230 C series sound and vibration input modules were integrated using a BNC connection, with the cRIO to read and condition the voltage signals from the accelerometers. Each NI 9230 module provided three input channels (used for *x*, *y* and *z* axes of each accelerometer), with sample rates up to 12,800 samples per second per channel, and a signal range of ±30 V.

Meggitt Endevco 65L-100 triaxial integrated electronics piezoelectric (IEPE) accelerometers were selected for the system as they have a wide frequency range and can effectively measure vibration from laboratory and industrial sources likely to be seen in measurement systems, i.e., acoustic vibrations (100 Hz to 10,000 Hz) and motorised equipment (20 Hz to 500 Hz). The sensitivity of 102.5 mV/ms$^{-2}$ is adequate to enable structural measurements in the micrometre amplitude region. Engineering systems may need to be condition monitored in six degrees of freedom, so triaxial accelerometers would only offer a solution to achieve measurement of vibrations in three degrees of freedom. However, the three degrees of freedom that accelerometers measure represent the largest vibration axes of interest in measurement systems, with torsional vibrations more relevant in applications with rotational inputs, such as the crank shaft in a combustion engine [20]. The accelerometers have two mounting options, which are adhesive mounting, where the bottom face is glued onto the frame, or screw mounted, and so offers a flexible solution that can be easily mounted on a variety of different measurement systems without requiring any design changes. The specifications of the accelerometers are given in Table 1. The accuracy and reliability of the system is assured as the performance specifications provided conform to ISA-RP-37.2 (1982) [21] and are typical values, referenced at +24 °C and 100 Hz, unless otherwise stated and the calibration is traceable to the National Institute of Standards and Technology (NIST).

**Table 1.** Endevco 65L-100 accelerometer specifications.

| Feature/Attribute | Specification |
|---|---|
| Model | 65L-100 |
| Sensitivity | 102.5 mV/ms$^{-2}$ |
| Number of axes | 3 |
| Mass | 5 g |
| Dimensions | (10 × 10 ×10) mm |
| Dynamic range | ±50 g |
| Measurement uncertainty | ±5% |
| Temperature range | −53 °C to +125 °C |
| Mounting | Adhesive or M2.5 thread |
| Frequency response | 1 Hz to 6000 Hz |

## 3. Software Framework

The software has been programmed to use the target and host architecture, which allows for the host computer to interact with the target virtual instrument (VI) compiled on the cRIO, while performing different operations in the host computer VI. The target VI is designed to collect data from the accelerometers and stream the data to three FIFOs (First-In-First-Out) buffers for the host VI to read. The host VI on the master computer is designed to read the data from the FIFOs in chunks, based on how much data is available when the read is requested, which will in turn depend on the sample rate and the execution time of the read loop. The host VI then performs the processing and analysis on the collected data, displaying the measurements on the front panel and streaming the data to disk. Utilising the cRIOs FPGA and the host–target architecture allows the system to achieve high sampling rates (up to 12,800 samples per second per channel), therefore, ensuring adherence to the Nyquist sampling theorem and measuring across the desired frequency range. High-level function breakdown diagrams of the target and host VIs architectures are shown in Figures 2 and 3, respectively.

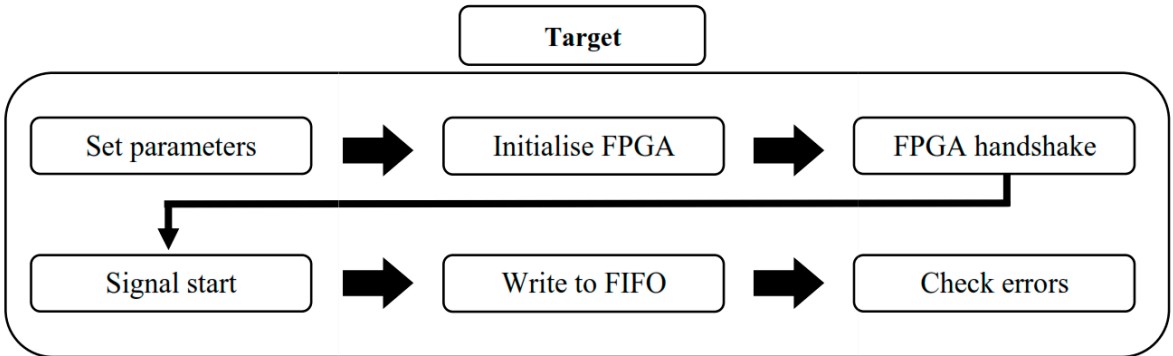

**Figure 2.** Programme function flowchart of the designed target code, which is compiled on board the cRIO, to collect and stream the accelerometer data to the host computer.

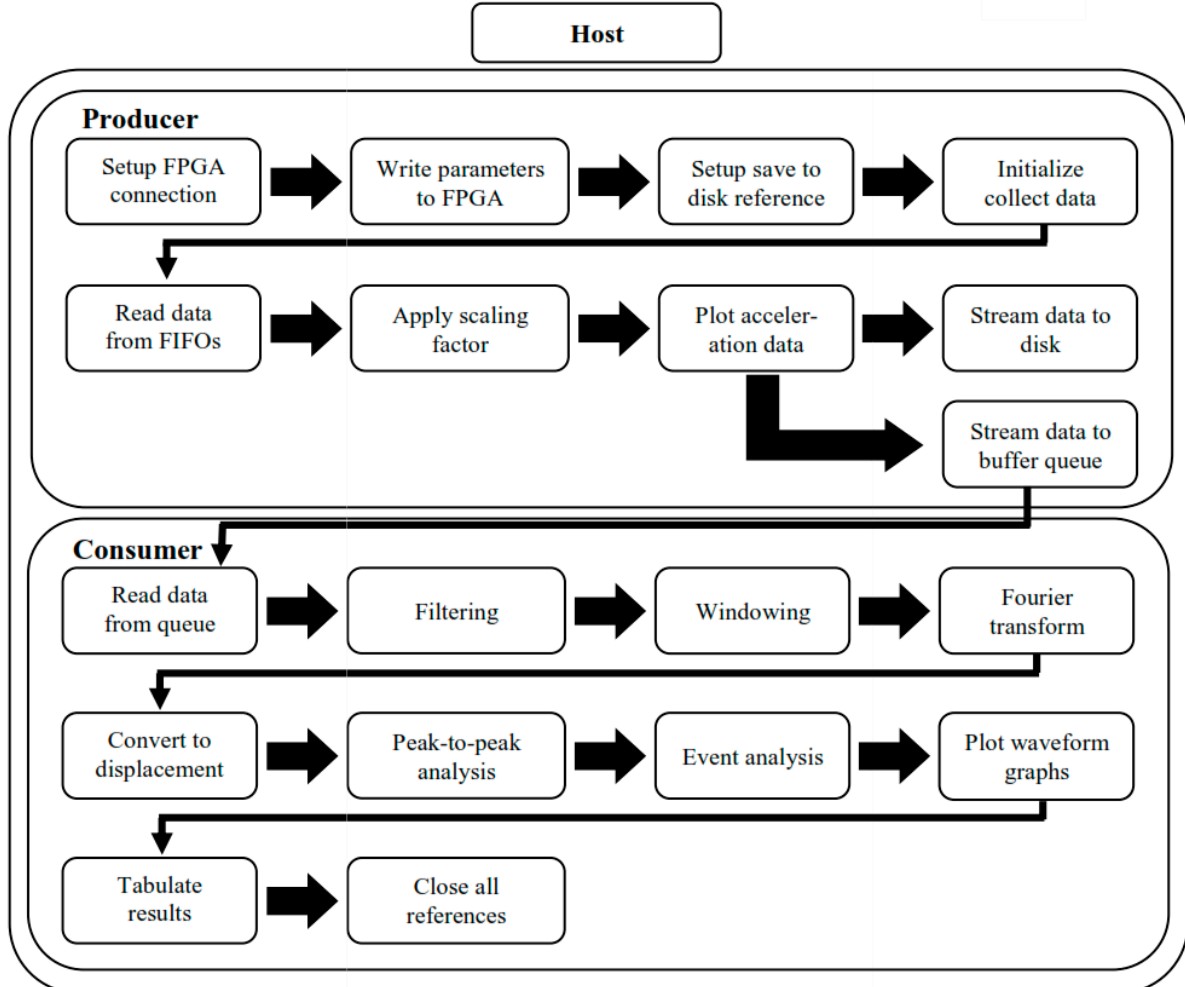

**Figure 3.** Programme function flowchart of the host script, which is executed on the main computer and communicates and reads the accelerometer data from the target VI on the cRIO.

### 3.1. Establishing Parameter Set

The settings and parameter section of the front panel allow the user to control key aspects of the device setup, measurement parameters and analysis settings. The cRIO device and input configuration are constants in the current accelerometer system, but could be readily adapted for integration of other accelerometer types. The sample rate can be easily changed to a number of available preset sample rates, ranging from 985 samples per second per channel to 12,800 samples per second per channel. The user defines the filename and file path for the measurement file so that the data can be streamed to disk. Key analysis settings can be set to specify the analysis time window, peak and valley threshold values, peak separation and type of windowing used. The analysis time window sets the length of time in seconds as decided by the user, i.e., the update rate of the results and window time of interest. Threshold values are used to determine the amplitudes above which the magnitudes of the peaks and valleys, used for the peak-to-peak analysis, are found. Peak separation sets the length of the moving window of points, which the peak finding function uses to search for peaks and valleys in the data. The system allows the user to choose the type of windowing applied to the data, to help produce clearer results during Fourier analysis of the data by reducing the effects of spectral leakage.

### 3.2. Filtering

As piezoelectric accelerometers are unreliable outside of their designed frequency range [22], it is necessary to filter the acquired signal to remove the unreliable and noisy data outside of the

measurement range. A second-order Butterworth filter [23] is applied to remove the unwanted frequencies, where the user defines the upper and lower ranges. A minimum of 5 Hz must be removed from minimum frequency range due to the unreliability of the accelerometer when measuring very low frequencies and as a result of the limits of the conversion method used to obtain velocity and displacement data.

### 3.3. Windowing Options

To analyse the frequency spectrum of the signal and convert the acceleration data to displacement data, the signal needs to be transformed into the Fourier space. To improve the clarity of the Fourier signal and reduce the effects of spectral leakage, window functions are commonly applied. Windowing reduces the amplitude of the discontinuities at the endpoints by multiplying the time signal by a finite length window function with an amplitude that smoothly reduces to zero at its endpoints, ensuring that the endpoints of the time signal meet and the signal forms a continuous waveform. The front panel of the host VI offers the user a choice of windowing function, which include Hanning, Hamming, Blackman, Blackman–Harris, Blackman exact and no window (often referred to as rectangular or uniform window) [24].

### 3.4. Signal Conversion

Many applications perform vibration analysis on the velocity and displacement behaviour in the system. To this end, the acceleration signal is converted to velocity and displacement using a technique called omega arithmetic [25], which provides significantly improved results in comparison to other available conversion techniques, such as the widely used double integration method, as it eliminates the errors from the integration constants by processing in the Fourier space. The omega arithmetic method derives the following relationships between acceleration, velocity and displacement:

$$\ddot{X}(f) = i\omega \dot{X}(f) \tag{1}$$

$$\ddot{X}(f) = -\omega^2 X(f) \tag{2}$$

where, $\omega = 2\pi f$, $\ddot{X}(f)$, $\dot{X}(f)$ and $X(f)$ are acceleration, velocity and displacement in the frequency domain respectively.

The omega arithmetic method has a low frequency limit of ~5 Hz, because as the frequency approaches zero, the $\dot{X}(f)$ derived from $\ddot{X}(f)$ becomes indeterminate; however, as the low frequency components are removed by filtering, this limit should not affect the velocity and displacement results in our measurement frame condition monitoring application.

### 3.5. Analysis Type

The type of analysis performed on the acquired signal is dependent on the application and the characteristics of interest in the signal. In the first instance, the system allows the user to choose the type of signal on which to perform the analysis, where the three main vibration signals that are measured are acceleration, velocity or displacement. Vibration analysis techniques can be separated into the different domains in which the analysis is performed, i.e., the time domain or the frequency domain.

The system utilises two main VIs for the processing and analysis of vibration data: one VI for data collection and live processing, and a second VI that performs further analysis on the complete measurement dataset. The live data VI has a limited processing time and RAM (~3.5 GB). The RAM is limited due to the real-time module, which is needed to develop and embed real-time applications on the cRIO, but currently only operates using a 32-bit processing system. Due to the processing time and RAM limitations, the amount of analysis and visual outputs in the live data VI is minimised to ensure the code can execute and update at the required rate. A data window strategy is used to provide data updates at regular time windows (every 2 s to 5 s). Displaying multiple graphics for eighteen

signals simultaneously is not practical with the current computational limits, and so an automatic peak detection algorithm has been developed to display the data for the three channels with the largest averaged peak-to-peak response over the previous measurement window. Updating the graphs using the peak response method allows the channels (position and axis) with the largest vibrations to be highlighted and displayed on the front panel, whilst ensuring the graph is still legible and the dataset is limited to a size, which can be updated in the processing time available.

To limit the amount of processing required in the live data VI, only a subset of the analysis options are available for the user to select on the front panel. The analysis options available in the live data VI are described below.

### 3.5.1. Peak-to-Peak

Peak-to-peak analysis looks at the distance from the positive peak to the negative peak (illustrated in Figure 4), which in the case of a simple sine wave would be twice the amplitude of the positive peak as the wave is symmetrical. Peak-to-peak analysis is useful when displacement is of interest, and ideal for analysing shock events or assessing clearances in for mechanical assemblies. For measurement system applications, the maximum displacements present during the measurement time are of particular interest, as the system will have a limit for the level of vibration displacement that the system can tolerate and still meet the specified measurement accuracy. The identified peaks and valleys that exceed the user defined threshold values are used to find the average peak-to-peak response of the measurement window.

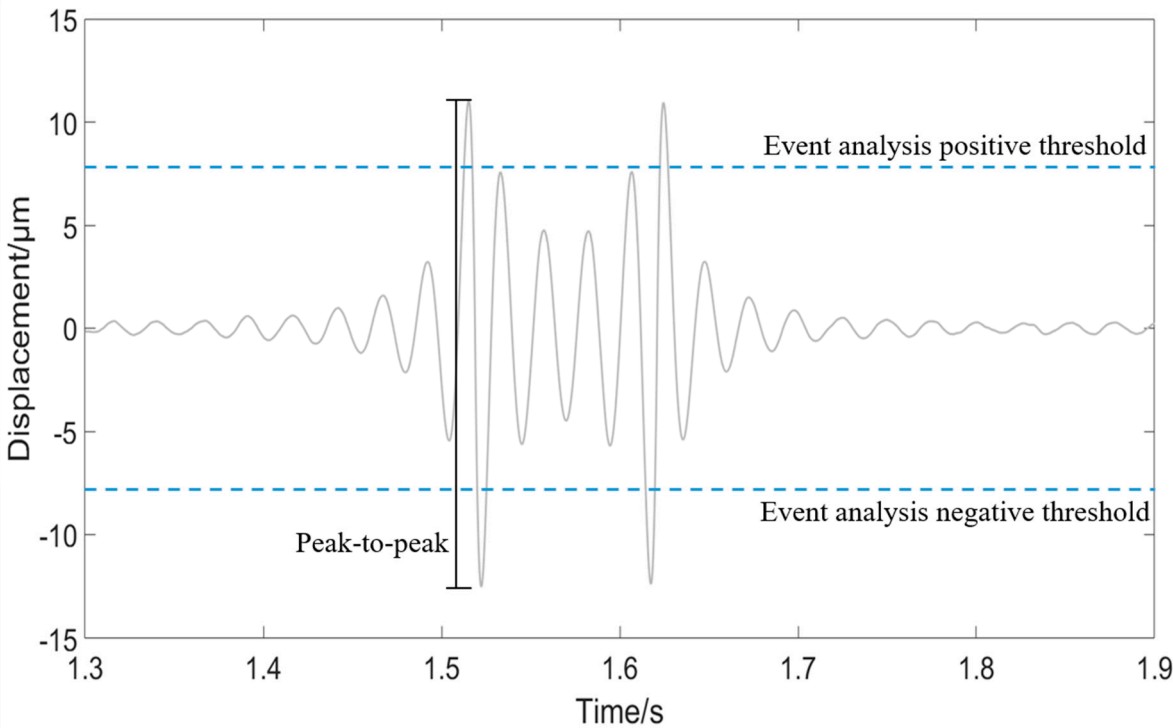

**Figure 4.** Event analysis plot to investigate the dynamic behaviour around the area of interest, which is identified by the peak above the user defined threshold (dashed line), with the peak-to-peak analysis also illustrated.

### 3.5.2. Frequency Spectrum Analysis

Frequency spectrum is an effective vibration analysis tool. By converting the signal into the Fourier domain using the fast Fourier transform (FFT) algorithm, the signal can be shown with respect to its vibration amplitude as a function of frequency. Analysing the frequency spectrum allows the different vibration signal contributors to be separated, so that the main excitation sources can be

identified. The resolution of the frequency (i.e., the size of each frequency bin) in an FFT is directly proportional to the signal length and sample rate, and so to improve the resolution, the number of samples used in the FFT must be increased.

### 3.5.3. Power Spectral Density

The power spectral density (PSD) is a measure of the signal's power content against frequency, and shows how the power of the signal varies with frequency. PSD is a useful analysis technique for comparison of random signals over different sampling lengths because the signals are normalised by the spectral resolution so the signals can be overlaid and compared.

### 3.5.4. Event Analysis

The post-measurement analysis VI is able to run on the 64-bit version of LabVIEW, and therefore does not have the same processing time and memory restrictions as the live data VI. As a result of the increased available RAM, the post-measurement analysis VI is able to perform analysis on much larger datasets, allowing signal analysis of the entire measurement length. In addition, as the processing time is not limited by the data collection loop execution, the post-measurement VI is able to provide additional analysis capabilities and displays such as event analysis and 3D stacked ribbon plots.

In condition monitoring, identification and analysis of vibration "events" is of particular interest, as sudden and sharp increases in vibration levels can be an indication that there is either a major fault within the system, or that an external source is interacting with the system. Event analysis identifies any significantly large peaks which are above the peak threshold values set by the user, and are much larger than the usual vibrations present. The "event" is plotted (Figure 4) with a much smaller time window to allow further investigation into the dynamic behaviour of the system before and after the vibration spike, i.e., to evaluate the dissipation time and check for any reoccurrence.

## 4. User Interface

The front panel (Figure 5) was designed for simplicity, so that the user can easily navigate and operate the system. The front panel is comprised of three main sections, which are the settings and parameters, the results and the system properties and errors.

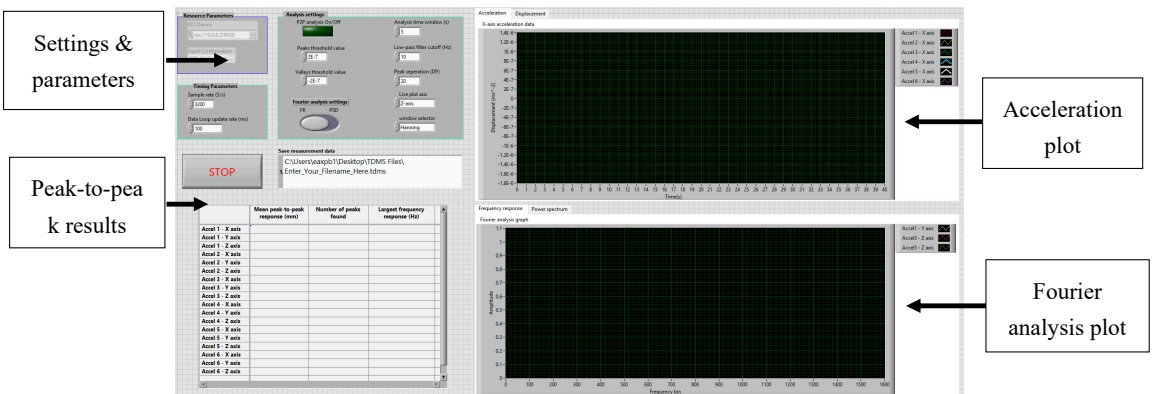

**Figure 5.** Front panel LabVIEW display of the multi-accelerometer host VI, which allows the user to control the system and view the measurement results.

### 4.1. Results—Live Data VI

The results section shows the raw acceleration data, which is updated live based on the user-chosen axis for all of the accelerometers; an example of acceleration results for the *z*-axis is shown in Figure 6. The results displayed update at the rate defined by the users chosen time window, with a minimum update window of 100 ms, which is the loop time of the data collection and processing loops. Viewing

the live data in the chosen axis for each position allows the user to see an immediate response of the vibration behaviour in the system. The second graph shows the results of the Fourier analysis, and shows a plot of the frequency response or the power spectral density (depending on the choice of the user). The displacement results for the three channels with the largest peak-to-peak responses are displayed on the graph, under the displacement tab on the front panel. The displacement plot updates at the rate of the user defined measurement window, and utilises an automatic peak detection algorithm to identify the channels with the largest responses from the previous measurement window. An example of the displacement results plot and frequency response can be seen in Figures 7 and 8, respectively. The average peak-to-peak response for all eighteen channels and the number of peaks found are tabulated on the front panel.

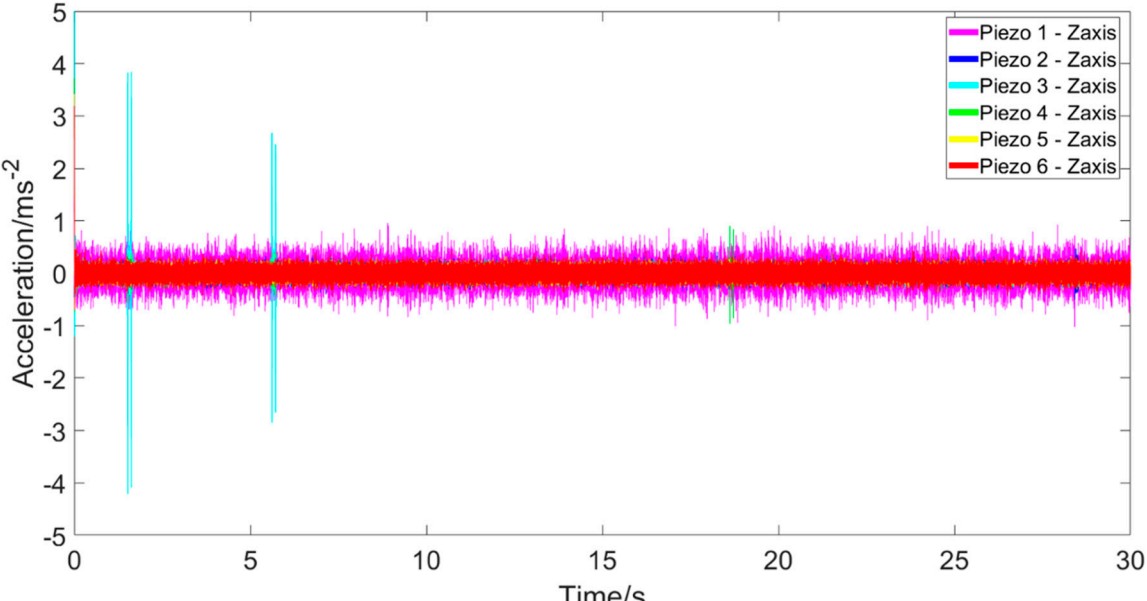

**Figure 6.** Live acceleration results for the *z*-axis for each of the six accelerometers allowing the user to immediately see any changes to the dynamics of the system.

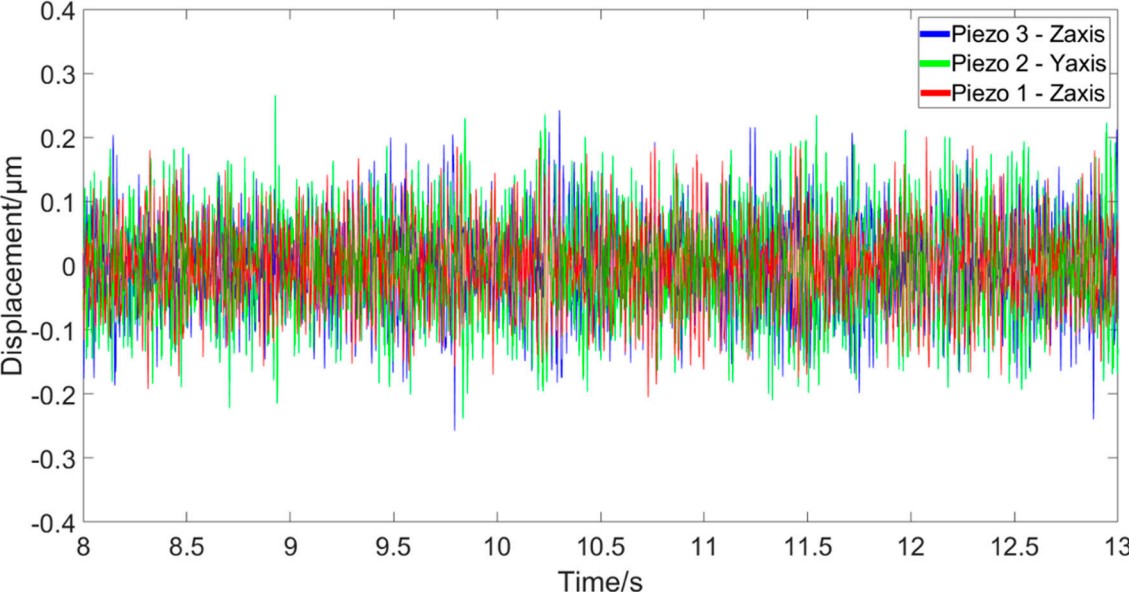

**Figure 7.** Displacement results for the three channels that have been identified to have the largest average responses in the system within the time window analysed, which was 5 s in the example shown.

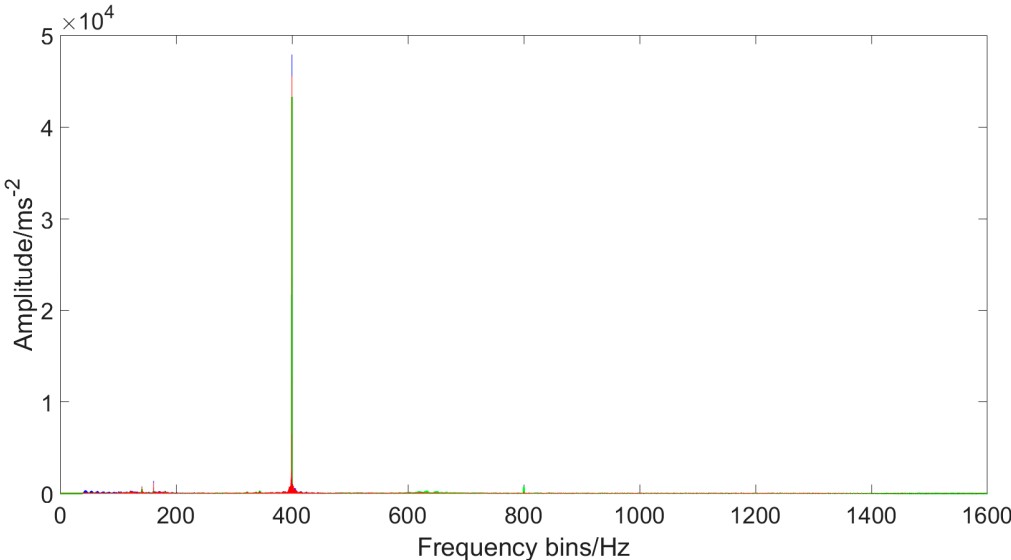

**Figure 8.** Frequency responses of the three channels identified to have the largest average responses in the system within the time window analysed, which was 5 s in the example and shows a dominant excitation input with a frequency of 400 Hz.

### 4.2. Results—Post-Measurement Analysis VI

The post-measurement analysis VI displays the displacement and frequency response for all eighteen measurement channels, with the signals separated into the axis of measurement. The post-measurement analysis processing time is dependent on the total time of the measurement, in the case of a 90 s measurement the processing time for the system is approximately 20 s. Given that displaying large datasets in multiple channels is challenging for vibration measurements, a 3D stacked ribbon plot (Figure 9) was created to show the displacement envelope of all eighteen channels on a singular plot to aid clear comparison. To make the plots distinguishable, the two-dimensional plots for each of the eighteen channels are stacked and evenly spaced along a third dimension and colour coded to help with visual identification. The 3D ribbon plot has also been used to highlight the location of the identified key "events". The 3D ribbon plot is currently not possible in the live data VI, due to the limits in RAM and processing time.

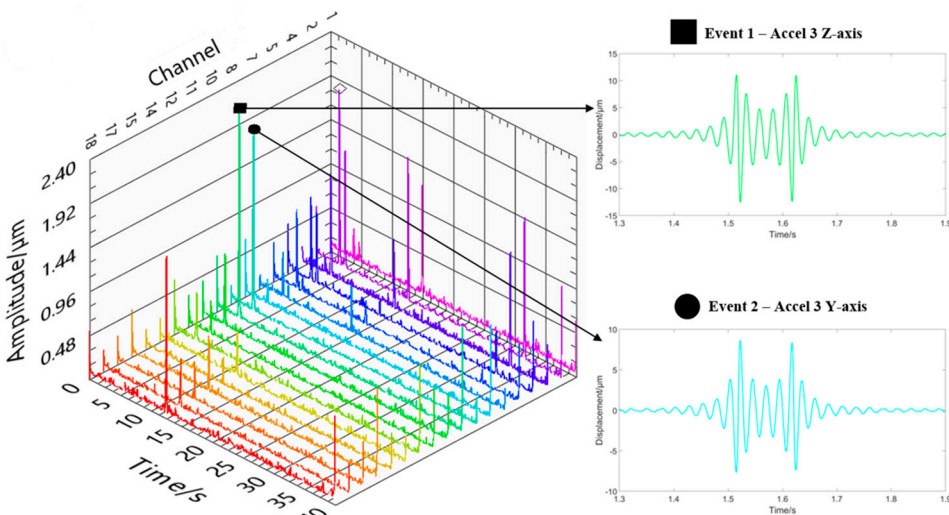

**Figure 9.** 3D stacked ribbon plot of the displacement envelope results for all 18 measurement signals ($x$, $y$ and $z$ axes for each accelerometer) with the key events that are identified to be of interest and so further analysed.

*4.3. System Properties and Errors Section*

The system properties and errors section displays key system performance parameters whilst the VIs are executing, including the elements remaining in each of the FIFOs and the buffer queue, so the user can check that they are all being properly emptied. The execution times of both the data collection and processing loops are also displayed, to check they are running at the required rates and to allow the user to identify in which loop the problem is occurring if either execution time increases. Finally, the LED indicators on the front panel that alert the user if either the FIFO or buffer queue are full. Error handling is an extremely useful tool for identifying and debugging any issues in the system; therefore, if the code has any runtime errors, they are displayed in the FPGA error out or the host error out sections at the bottom of the system properties and errors section, depending on which part of the project has experienced the error.

*4.4. Data Handling*

In the live data VI the raw accelerometer data is converted from raw voltage data to acceleration data using the sensors sensitivity factor, and then streamed to a TDMS file. This allows for the data to be processed live to monitor the performance/condition of the system, and then allow for post process analysis to be carried out on the complete measurement dataset.

Another important step is presenting the processed results in a clear and user-friendly manner, so the front panel has been optimised to ensure the user can navigate and follow the information easily, where the quantity of information displayed is limited to ensure simple interpretation for the user. To supplement this, an automatic report generation tool was designed to present all of the data and analysis performed in a clear and easy to read report. As well as including all the key results and data, the report includes details of the parameters and settings used during measurement (e.g., sampling rate, measurement time, number of accelerometers and lab temperature) and an explanation of the methods used for analysis (including filtering and windowing applied during data processing), providing a useful account for reference and interpretation.

## 5. Lathe Condition Monitoring Investigation

To demonstrate the flexibility of the system and highlight the analysis tools an experimental investigation into the effects of cutting depth and tool wear was performed on a typical CNC lathe to show the effectiveness of the developed conditioning monitoring system. Three arbitrary locations were identified to mount the accelerometers on the lathe to allow the different components of the system to be monitored during operation, with the accelerometers readily integrated on the lathe using an adhesive. The locations (shown in Figure 10) were the tailstock, cutting bed and the headstock.

To investigate the dynamic effects of different cut depths on the condition of the system, the vibrations in three axis were measured for a range of 0.5 mm to 4.0 mm cutting depths with steps of 0.5 mm. A cut of length 100 mm was performed for each cutting depth, with a new piece of stock material used for each test. The experiment was performed once with a new cutting tip, and once using a "worn" cutting tip (shown in Figure 11), which had been used for 40 h to perform light to moderate cutting operations from a previous part, but was deemed in a "usable" condition. The cutting tips used were CP200 carbide with a TiAlN/TiN finish, diamond, 1/4″ thick, angle of 55°, and 1/32″ corner radius. A number of parameters were set throughout the experiment and have been shown in Table 2.

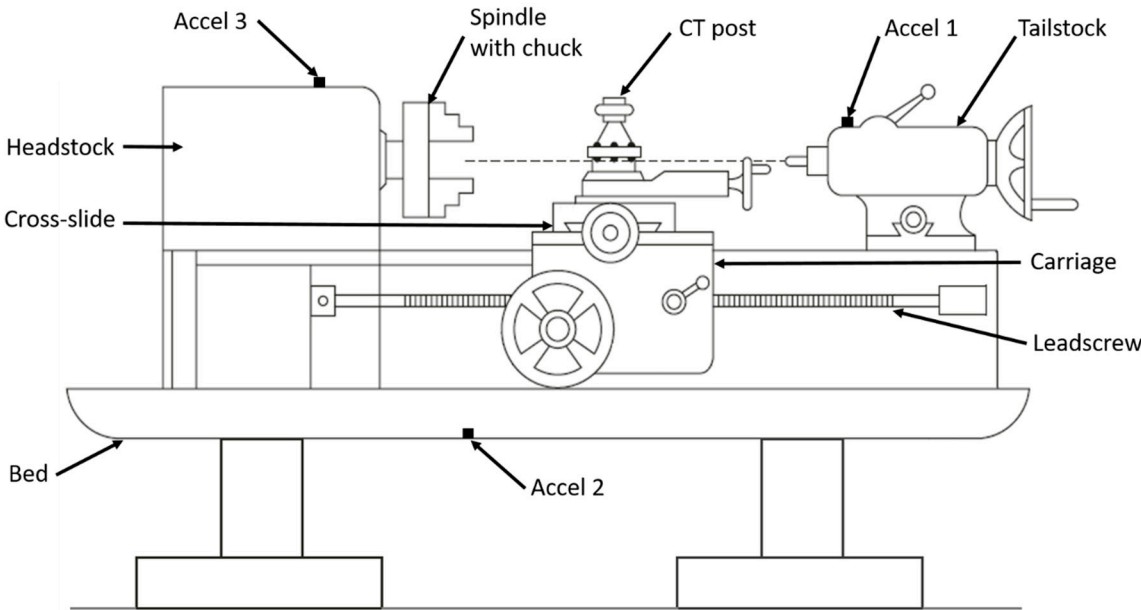

**Figure 10.** Schema of the experimental configuration showing the key components of a lathe, and the accelerometer mounting positions for the condition monitoring investigation into the effects of cutting depth and tool wear.

**Table 2.** CNC lathe settings used in the experimental condition monitoring experiments.

| Attribute | Setting |
| --- | --- |
| Spindle speed | 600 RPM |
| Feed rate | 0.15 mm/rev |
| Cut length | 100 mm |
| Material | EN 24 steel |
| Diameter | 38 mm |

**New** **Worn**

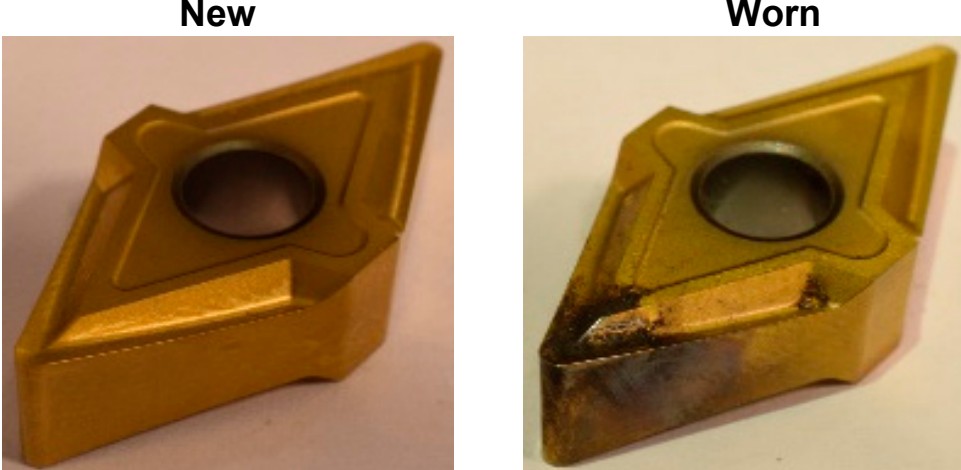

**Figure 11.** Visual representation of the new cutting tip (**left**) and worn cutting tip (**right**) used for the lathe tool wear investigation.

An example of the measured vibration signal for a 2 mm cutting depth using the new cutting tool can be seen in Figure 12, and the results of the frequency response analysis in Figure 13. The frequency analysis identified the largest frequency components of all three measurement positions to be at ~10 Hz, which matches well with the vibrations that would be introduced to the system due to the spindle rotating at 600 RPM. The frequency analysis also showed the main frequency components

in the lathe system are between 8 Hz to 50 Hz. From each of the measured signals, the average peak-to-peak response was calculated for the cutting process. The results for the $x$, $y$, and $z$ axes from the investigation using both new and worn cutting tips can be seen in Figure 14.

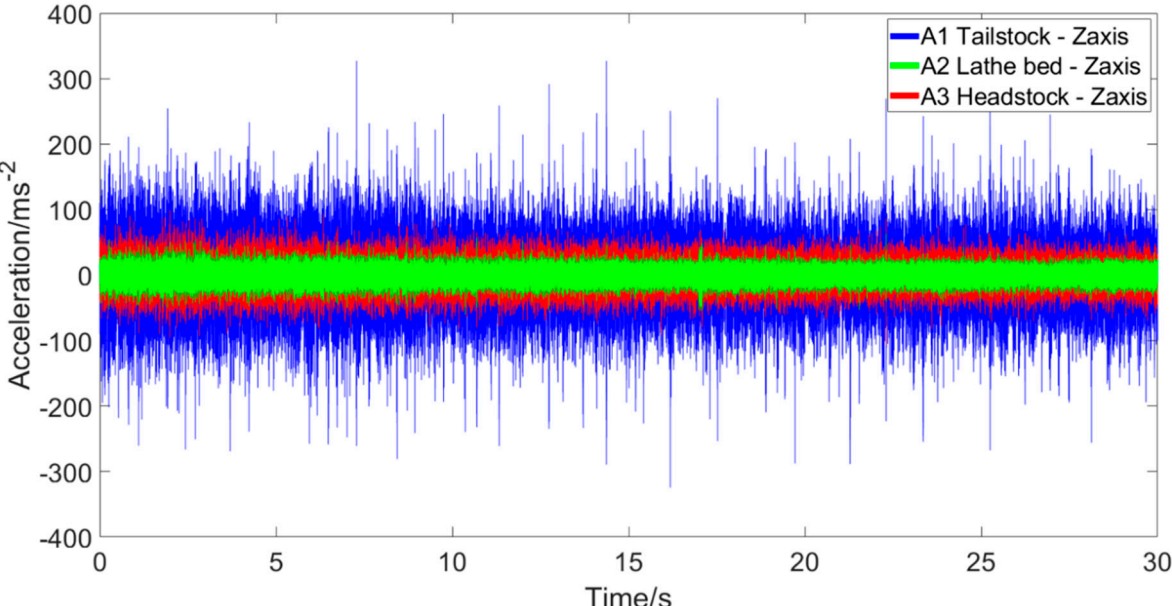

**Figure 12.** Example measured vibration dataset, collected while performing a 2 mm depth of cut on a CNC lathe using a new cutting tip.

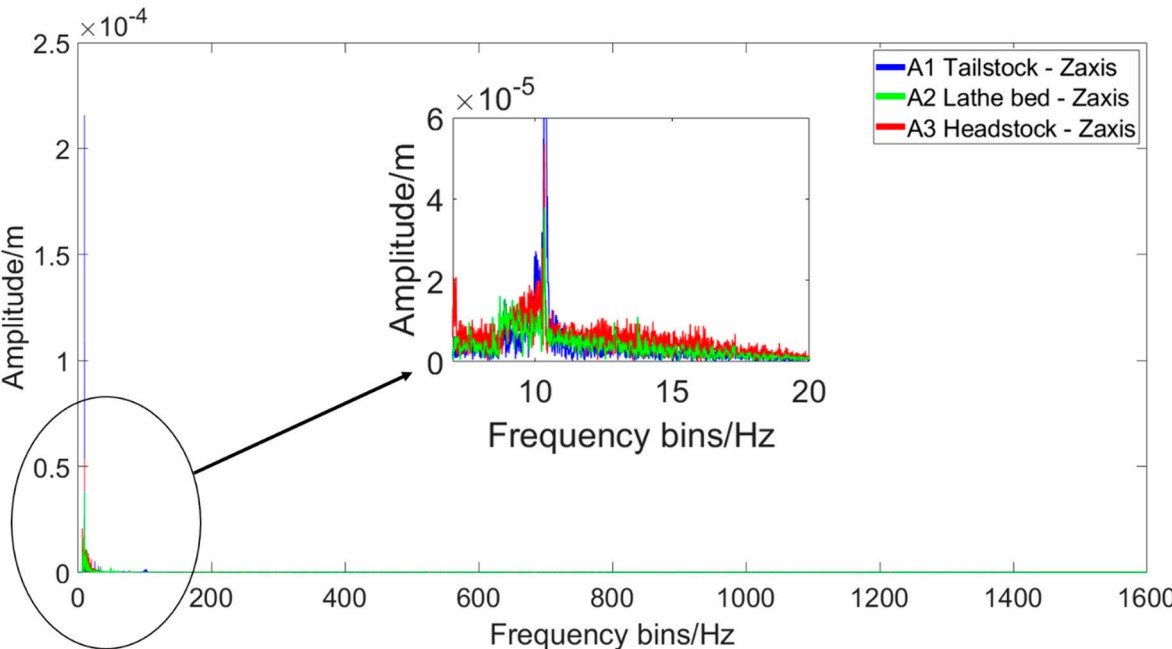

**Figure 13.** Example frequency response analysis plot of the measured vibration dataset collected whilst performing a 2 mm depth of cut on a CNC lathe using a new cutting tip, showing the largest response at 10 Hz, which matches the frequency of the spindle.

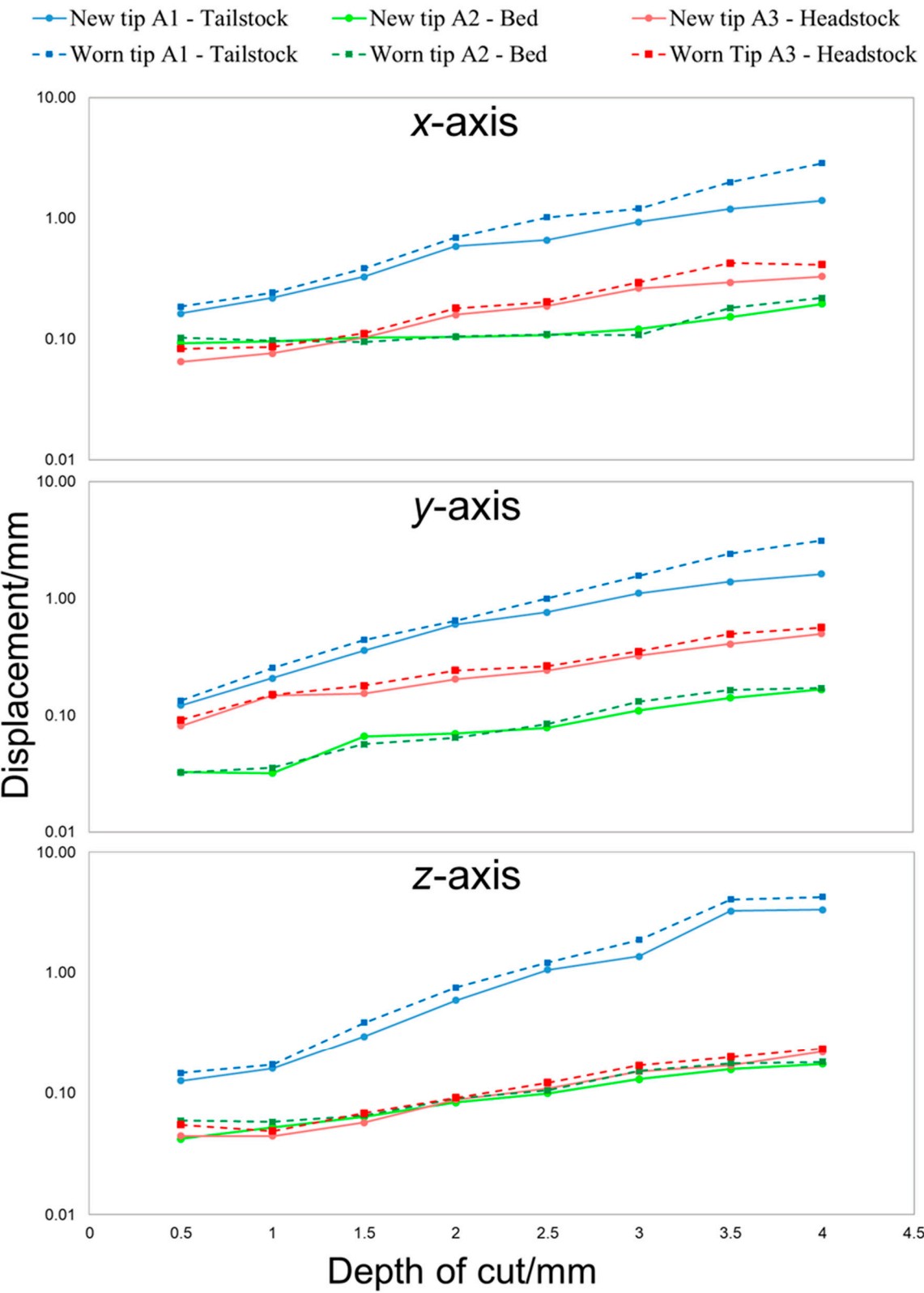

**Figure 14.** Vibration results in the *x*, *y* and *z* axes for the effect of increasing cutting depth and tool wear, showing the vibration increases that can be measured as the cutting tip progressively wears to predict tool failure.

It is clear from the results that a larger cutting depth causes an increase in vibration in all three axes, and is consistent in all three positions measured. The largest variation in vibrations was in

the tailstock, as it is directly connected to the work piece, and so experiences significantly increased vibration from the cutting process and the spindle rotating. The tailstock results, showed there was a marked rise in the average peak-to-peak vibration response when performing cutting operations with a worn tip compared to the new tip in all three measured axes. The trend in both the new and worn cutting tip are similar with increasing cutting depth, but there is a clear upward shift in the worn cutting tip results; this is a clear indicator of the effects of wear on tip. The clear detection of the increase in vibration due to tool wear in the tailstock demonstrates the effectiveness of the developed condition monitoring system, and shows it offers a flexible method to verify the vibration performance in a variety of applications. In the presented study of lathe cutting, further testing would allow for a vibration performance metric to be defined, which could be used to quantify the threshold of acceptable vibrations in order to monitor the condition of the cutting tip and predict tool tip failures before they occur. An example plot generated based on event analysis function of the developed system has been shown in Figure 15.

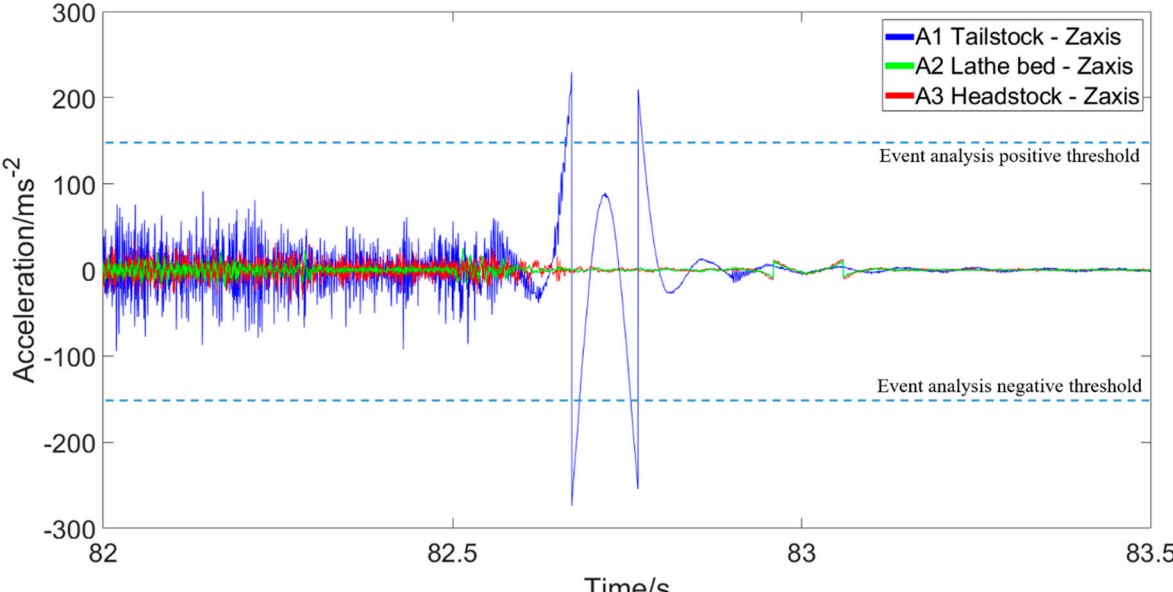

**Figure 15.** Example of an "event" identified by the event analysis tool, as the measured signal exceeds the user defined thresholds (dashed lines). In this example, the "event" is as a result of the brake being applied to stop the spindle after the cutting process has finished.

The "event" identified in Figure 15 highlights the event analysis function of the developed system, and alerts the user to further investigate the behaviour as an area of interest, due to the higher than "normal" amplitude of the vibrations measured. In the example shown, after further investigation the event was identified as due to the brake being applied to stop the spindle after the cutting process had finished. The lathe investigation demonstrated the use of the developed multipoint condition monitoring system, and shows how effective condition monitoring can enhance tool wear detection in machine applications, by measuring the increase in vibrations due to the wearing of the cutting tip. The developed system has clearly shown the functionality to readily measure and analysis the vibration changes in the structure to provide this condition monitoring capability.

## 6. Conclusions

A flexible experimental multipoint vibration analysis performance monitoring and characterisation system has been detailed. The system offers a flexible solution for a variety of applications, and can characterise the dynamics of a structure in $x$, $y$ and $z$ axes, with a sensitivity of 102.5 mV/ms$^{-2}$, data acquisition (up to 12,800 samples per second per channel) and a frequency range of 5 Hz to 6000

Hz. The developed system can rapidly process the collected dynamic performance data, and offers online condition monitoring capabilities to evaluate the structural performance live, with further post measurement analysis functions being available. An experimental investigation into the effects of cutting depth and tool wear was performed on a CNC lathe, to demonstrate the capabilities of the developed system and the benefits of effective condition monitoring in a machine application. The system has been developed to be easy to implement to any structure, with a range of analysis functions so has the potential to be used across a wide range of applications. The flexibility of the system has been demonstrated by performing a lathe cutting tool investigation, this highlighted the readiness of the system to be used in a range of applications.

**Author Contributions:** In the work presented in this paper the contributions to each section of the work was as follows. Conceptualisation: P.B., A.C. and R.L.; methodology: P.B.; software: P.B. and L.T.; validation: P.B. and L.T.; formal analysis: P.B.; investigation: P.B.; resources: P.B., A.C. and R.L.; data curation: P.B.; writing (original draft preparation): P.B.; writing—review and editing: P.B., L.T., A.C. and R.L.; visualisation: P.B.; supervision: A.C. and R.L.; project administration: P.B., A.C. and R.L.; funding acquisition: R.L. All authors have read and agreed to the published version of the manuscript.

**Funding:** This research was funded by the EPSRC grants EP/M008983/1 and EP/L016567/1.

**Acknowledgments:** The authors would like to acknowledge the support provided by Alex Jackson-Crisp and Simon Lawes (University of Nottingham).

**Conflicts of Interest:** The authors declare no conflict of interest.

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
