# Peer review of "Performance Verification of a Flexible Vibration Monitoring System"

_machines, doi:10.3390/machines8010003_

Round 1
Reviewer 1 Report
This paper introduced a flexible experimental multi-point vibration analysis performance monitoring and characterisation system. The author elaborated the software and hardware of the system, as well as the programme function of the software. Whilst the signal acquation, several types analysis including peak-to-peak, power spectral density and event analysis can be performed. The graphic interface is user-friendly, and can present the graphic results in real-time. An experimental investigation into the effects of cutting depth and tool wear was performed on a CNC lathe to verify the capabilities of the developed system and the result proved to satisfactory.
However, there are several problems in the content of the article and the overall design of the system. The following suggestions and questions are proposed for some deficiencies in this paper
Suggestions:
In section 2, the author introduces the software and hardware of the system, and the programme function flowchart is drawn. But there are no images to introduce the hardware, so it is suggested to place a physical picture in the paper to express the hardware and illustrate their connection. The format of equation (1) and (2) is recommended to be modified. Figure 12-14 show the vibration results in x, y and z axes for the effect of increasing cutting depth and tool wear but there's some repetitive content such as images of the lathe and cutting tip on the right. It is recommended to give a hybrid graph to demonstrate the displacement in 3 axes along with the depth of cut, and the three simplified curve diagram can be placed under that. Can the author give an example of the generated report? How to evaluate the effectiveness and accuracy of measurement results? The author claims the developed system can achieve the real-time performance monitoring. Is there any time-delay while the analysis of live data VI? How long did the post-measurement analysis VI take to display the displacement and frequency response for all eighteen measurement channels in the experimental study?
Author Response
Thank you for your review. Please see attachment.

Reviewer 2 Report
The manuscript presents a measurements system using accelerometers for condition monitoring focusing on hardware/software architecture. The manuscript is well-written and presented, but according to the reviewer does no present any novel concept or method.
The following comments are given:
1) Authors state at line 50: "we have developed a novel condition monitoring system", but the novelty should be better highlighted. According to the reviewer hardware is commercial, software architecture is not innovative and analysis is standard.
2) Authors should include additional information on signal acquisition and processing, e.g., filter characteristics (order).
3) Section 5, have repetitions been carried out on the experiments? The differences between worn and new tool should be discussed considering also bar errors and confidential interval.
4) A picture of the lathe with accelerometers installed should be added.
Author Response

(The authors gave the same response as above.)

Reviewer 3 Report
This paper presents a scientifically valuable and up-to-date study regarding the development of a novel condition monitoring system. The paper can be accepted for publication after some minor revision. The detailed remarks are:
despite the Introduction section presents the state-of-the-art regarding the characterization of measurement systems based mainly on accelerometers, it misses some works regarding the application of laser displacement and vibrometry measurement systems during machining operations. Therefore, it is advised to fill this gap and extend the literature survey in relation to the following works (among others):
1.Measurement of milling tool vibrations during cutting using laser vibrometry. International Journal of Machine Tools & Manufacture 2008;48:380–7. 2.Machined surface roughness including cutter displacements in milling of hardened steel. Metrology and Measurement Systems 2011;XVIII:429–40. 3.Effect of tool stiffness upon tool wear in high spindle speed milling using small ball end mill. Precision Engineering 2001;25:145–54.
In section "5. Lathe condition monitoring investigation", please provide the details regarding the geometry of the applied tool (e.g. corner radius, flank and rake angles), since it has direct effect on the dynamics of the machining process. It also advised to express some wear indicators (as the flank wear width, crater on the rake face, etc.) of the ‘worn’ cutting tool.
Author Response

(The authors gave the same response as above.)

Reviewer 4 Report
Congratulations to the authors for their good work.
Author Response
Thank you for your review. No changes were required.
Round 2
Reviewer 1 Report
This paper introduced a flexible experimental multi-point vibration analysis performance monitoring and characterisation system. The author elaborated the software and hardware of the system, as well as the programme function of the software. Whilst the signal acquation, several types analysis including peak-to-peak, power spectral density and event analysis can be performed. The graphic interface is user-friendly, and can present the graphic results in real-time. An experimental investigation into the effects of cutting depth and tool wear was performed on a CNC lathe to verify the capabilities of the developed system and the result proved to satisfactory.
The paper was well organized, the author has corrected the problems and suggestions raised earlier. Above all, the author had a thorough analysis of the research and development status of the articles involved in the field. The system has been developed to be easy to implement to any structure, with a range of analysis functions, so has the potential to be used across a wide range of applications. The article was fluent and the research had a certain degree of practicality.
Author Response
No changes required.
Reviewer 2 Report
Authors provide a rebuttal to the comments, I am still concern about the novelty of the manuscript, however now this aspect is clarified in the introduction. Moreover, authors stated that "The purpose of the lathe investigation was to demonstrate the effectiveness of the development condition monitoring system and show the flexibility to be readily used in a variety of applications." this should be highlighted also in the manuscript.
Author Response
The flexibility of the system is an important feature of the developed condition monitoring system and so should be highlighted more clearly. To address this comment, a discussion about the flexibility of the system has been added to the conclusions section, which highlights that the system was designed to be flexible for a wide range of applications, and this feature has been demonstrated through the lathe tool wear invetigation presented in the paper.